# Increasing Neuroinflammation Relates to Increasing Neurodegeneration in People with HIV

**DOI:** 10.3390/v15091835

**Published:** 2023-08-30

**Authors:** Azin Tavasoli, Benjamin B. Gelman, Christina M. Marra, David B. Clifford, Jennifer E. Iudicello, Leah H. Rubin, Scott L. Letendre, Bin Tang, Ronald J. Ellis

**Affiliations:** 1Department of Neurosciences, University of California San Diego, San Diego, CA 92093, USA; atavasoli@health.ucsd.edu; 2Department of Pathology, University of Texas at Galveston, Galveston, TX 77555, USA; bgelman@utmb.edu; 3Department of Medicine, University of Washington, Seattle, WA 98195, USA; cmarra@u.washington.edu; 4Department of Neurology, Washington University in St. Louis, St. Louis, MO 63130, USA; clifforddb@wustl.edu; 5Department of Psychiatry, University of California San Diego, San Diego, CA 92093, USA; jiudicello@health.ucsd.edu (J.E.I.); bit001@health.ucsd.edu (B.T.); 6Department of Molecular and Comparative Pathobiology, The Johns Hopkins University, Baltimore, MD 21218, USA; lrubin@jhu.edu; 7Department of Medicine, University of California San Diego, San Diego, CA 92093, USA; sletendre@ucsd.edu

**Keywords:** HIV, viral suppression, neuroinflammation, neurodegeneration, inflammatory biomarkers

## Abstract

Background: HIV infection causes neuroinflammation and immune activation (NIIA) and systemic inflammation and immune activation (SIIA), which in turn drive neurodegeneration (ND). Cross-sectionally, higher levels of NIIA biomarkers correlate with increased biomarkers of ND. A more convincing confirmation would be a longitudinal demonstration. Methods: PWH in the US multisite CHARTER Aging project were assessed at a baseline visit and after 12 years using standardized evaluations. We measured a panel of 14 biomarkers of NIIA, SIIA, and ND in plasma and CSF at two time points and calculated changes from baseline to the 12-year visit. Factor analysis yielded simplified indices of NIIA, SIIA, and ND. Results: The CSF NIIA factor analysis yielded Factor1 loading on soluble tumor necrosis factor type-2 (sTNFR-II) and neopterin, and Factor2, loading on MCP1, soluble CD14, and IL-6. The SIIA factor analysis yielded Factor1 loading on CRP, D-dimer, and Neopterin; Factor2 loading on sTNFR-II. The ND analysis yielded Factor1 loading on Phosphorylated tau (p-tau) and Aβ42; Factor2 loading on NFL. NIIA Factor1, but not Factor2, correlated with increases in CSF NFL (r = 0.370, *p* = 0.0002). Conclusions: Increases in NIIA and SIIA in PWH were associated with corresponding increases in ND, suggesting that reducing neuro/systemic inflammation might slow or reverse neurodegeneration.

## 1. Introduction

The human immunodeficiency virus (HIV) targets and systematically undermines the immune system’s vital components, thereby compromising the host’s ability to mount effective defenses against various pathogens [1]. Over 40 million people are infected by HIV globally [2]. People with HIV (PWH) have increased lifespans thanks to the widespread adoption and advancements in combined antiretroviral therapy (cART). In 1996, the life expectancy of those with HIV receiving cART was only 55 years, but now it resembles that of the general population [3,4]. CD4 plays a critical role in HIV infection by serving as the initial receptor that allows the virus to attach to and enter target immune cells. The progressive loss of CD4+ T cells due to HIV infection leads to severe immunosuppression and a compromised immune response, the underlying cause of opportunistic infections and diseases associated with acquired immune deficiency syndrome (AIDS) [5]. Although HIV is known for its damaging effects on the immune system, which ultimately lead to AIDS, HIV can also cause neurological complications, including HIV-associated neurocognitive disorders (HAND) [6]. HIV can induce chronic inflammation within the brain, characterized by the activation of immune cells. This ongoing neuroinflammation contributes to neuronal damage and degeneration over time [7]. HIV infection triggers an inflammatory response that disrupts the equilibrium of chemokines in both plasma and cerebrospinal fluid (CSF). This alteration affects immune cell migration and communication [8,9,10,11]. Cytokines serve as immunomodulatory proteins, directing immune cell recruitment to infection sites and governing their activities. Moreover, they possess the ability to impact the functionality and structural integrity of various cell types, including neurons, extending their influence beyond the immune domain. This dual role highlights the potential for cytokines to affect immune–neural interactions [12]. Since most neural cells express cytokine receptors, cytokines can affect neuronal signaling and repair, potentially disrupting cascades that underlie information processing, synaptic plasticity, and neurotransmitter release. Furthermore, the ability of cytokines to elicit cellular responses within neurons might either exacerbate or attenuate the reparative processes following neural injury or insult [11].

The virus infiltrates the brain in the initial stages of primary infection, within the initial 14 days [13]. Even at this early stage, HIV establishes latent reservoirs within microglia and CNS macrophages. These cells become sanctuaries for HIV, evading immune surveillance and potentially leading to reactivation when ART is discontinued [14]. Activated CD4+ T cells in lymphoid tissue play a central role in producing and releasing virions, which contribute to the systemic viral load detected in patient plasma. This quantification reflects the ongoing interplay between viral replication and the host immune response, offering insights into disease progression and dynamics [15]. The virus detected in CSF reflects systemic and CNS reservoir contributions [16] The virus found in CSF reflects a balance influenced by the virus coming from active CD4+ T cells in lymphoid tissue, the whole body’s circulation, and the hidden places in the brain where cells like microglia and CNS macrophages live [16,17]. The extensive utilization and effectiveness of ART have fundamentally transformed both the clinical progression of HIV disease and the prevalence of neurodegeneration [18]; it allows PWH to survive the disease, which exposes neurons to persistent inflammation, contributing to neurodegeneration and neurocognitive impairment [18,19]. Inflammation can be evaluated by measuring specific biomarkers in both plasma and CSF. These biomarkers provide insights into immune responses, aiding in disease understanding, prognosis, and treatment monitoring [20]. For example, neopterin, a product of the guanosine triphosphate pathway, is synthesized in monocyte/macrophage cells. It serves as a marker of immune activation, offering insights into cellular dynamics. Neopterin levels correlate with reactive oxygen species (ROS) released from macrophages, potentially relevant to neurodegeneration. Neopterin also triggers proinflammatory signaling, contributing to immune activation and inflammation. This interplay offers insights into both neurodegenerative processes and immune modulation [21]. Even though plasma and CSF HIV RNA levels have been effectively lowered to levels undetectable by clinical assays (<50 copies of HIV RNA/mL), there is often a sustained mild elevation in CSF neopterin. This indicates the presence of persistent low-level immune activation within the central nervous system [21].

HIV gene products, such as gp120 and Nef, stimulate the activation of lymphocytes and macrophages, resulting in the secretion of proinflammatory cytokines and chemokines [22]. HIV proteins imitate and enhance TNF-receptor signaling, causing persistent HIV replication in infected cells through activation of nuclear factor (NF)-κB, a prototypical proinflammatory signaling pathway [23]. Soluble tumor necrosis factor receptor II (sTNFR-II) holds a significant position within the intricate network of immune signaling. It serves as the extracellular counterpart to the cell membrane-bound receptor for tumor necrosis factor-α (TNF-α), a potent cytokine predominantly secreted by macrophages. This multifunctional receptor extends its reach beyond TNF-α alone, as it also intricately interacts with lymphotoxin-α, a distinct signaling molecule primarily secreted by T lymphocytes and NK cells [24]. Elevated pretreatment levels of sTNFR-II in plasma hold profound implications within the context of HIV infection dynamics and the therapeutic landscape. This biochemical marker, situated at the crossroads of immune modulation, emerges as an important clue of disease progression and mortality in individuals undergoing ART. The link between increased pretreatment levels of plasma sTNFR-II and heightened mortality underscores the receptor’s potential as a prognostic marker [25]. TNF- α -induced damage to the deep white matter contributes to changes in cognition, while the brain’s response to HIV infection also plays a role in the onset of HIV-associated dementia. This occurs through widespread immune activation triggered by cytokines [26].

Neurodegeneration is often assessed by measuring biomarkers of neuronal injury in plasma and CSF. The most frequently used biomarker of neurodegeneration is the neurofilament light protein (NFL), a major structural element of large-caliber myelinated axons. CSF NFL concentrations represent a sensitive marker of damage to both central and peripheral neurons in several neurologic disorders [27,28,29,30,31,32]. In PWH, CSF NFL levels are negatively correlated with blood CD4+ nadir T lymphocyte counts, demonstrating the relationship between neuronal injury and systemic HIV infection [33].

A comprehensive exploration of these neurological consequences is essential not only to deepen our understanding of the virus’s multifaceted impact but also to inform the development of integrated therapeutic strategies that address both immunological and neurological dimensions of HIV infection. We measured changes in biomarker levels from baseline to follow-up over 12 years, hypothesizing that increases in inflammation would correlate with increases in neurodegeneration biomarkers. We also expected that viral suppression between the baseline and follow-up visits would be associated with decreasing markers of neuroinflammation and immune activation (NIIA) and systemic inflammation and immune activation (SIIA).

## 2. Materials and Methods

### 2.1. Participants

PWH in the US multisite CNS HIV Antiretroviral Effects Research (CHARTER) Aging project were assessed at a baseline visit and again after 12 years using standardized evaluations. Baseline visits took place from 2003–2007; follow-up visits were performed between 2016 and 2019. Participants with severely confounding medical and neuropsychiatric conditions such as active psychosis were excluded. All participants signed informed consent documents approved by the local Institutional Review Boards at each site.

Briefly, the standardized assessment included a medical history, psychiatric interviews to obtain Diagnostic and Statistical Manual Version IV (DSM-IV) diagnoses, neurological and physical examinations, and laboratory assessment as previously described [34]. Participants were evaluated by trained clinicians using standardized clinical examinations as previously described [35]. Comprehensive medical history including demographics and current and past exposure to specific ART drugs has been completed. Blood was collected via phlebotomy and CSF via lumbar puncture. CD4+ T-cell count was measured by flow cytometry in Clinical Laboratory Improvement Amendments (CLIA)-certified laboratories at each site. The quantification of HIV RNA levels, a crucial indicator of viral activity, was carried out in both CSF and plasma using real-time polymerase chain reaction (PCR) technology with a lower quantification limit of 50 copies per milliliter. Potential participants were included in this analysis if their plasma HIV RNA was ≤50 copies/mL at follow-up and data from a panel of 14 soluble biomarkers measured by immunoassay were available at both assessments: interleukin 6 (IL-6; CSF and plasma), soluble tumor necrosis factor type II (sTNFR-II; CSF and plasma), neopterin (CSF and plasma), monocyte chemoattractant protein type 1 (MCP-1; CSF and plasma), soluble CD14 (sCD14; CSF and plasma), and 8-Oxo-2′-deoxyguanosine (8-oxo-DG—a marker of nucleic acid oxidation; CSF and plasma), neurofilament light (NFL; CSF only), total tau (CSF only), phosphorylated tau (CSF only), soluble amyloid precursor protein (sAPP)-α (CSF and plasma) and amyloid β-42 (Aβ42; CSF only), soluble CD40 ligand (sCD40L; plasma only), C-reactive protein (CRP; plasma only), and D-dimer (plasma only).

### 2.2. Statistics

Demographics and clinical characteristics were summarized using numbers and percentages, means and standard deviations, and medians and interquartile ranges (IQR). For each biomarker available, we calculated the difference between its levels at the baseline and follow-up visit. Factor analyses were used to construct simplified indices of biomarker changes separately for biomarkers of SIIA (plasma sTNFR-II, neopterin, MCP-1, sCD14, sCD40L, and CRP, D-dimer, and IL-6), NIIA (CSF sTNFR-II, neopterin, MCP-1, sCD14, and IL-6), and ND (CSF NFL, total tau, p-Tau, sAPP-α, Aβ42 and 8-oxo-DG). Correlations of SIIA and NIIA with ND indices were evaluated using Pearson’s r or Spearman’s rho as appropriate. Follow-up analyses assessed correlations between the individual biomarkers separately. Assessment of potential confounds including demographics and indicators of HIV disease status was done using multiple regression.

## 3. Results

Participants were 108 ART-treated PWH, all virally suppressed at follow-up, with demographics and HIV disease characteristics as shown in Table 1.

Baseline visits occurred between September 2003 and September 2008; follow-up visits occurred between March 2016 and June 2019. The mean duration between visits was 12.5 ± 0.766 years. Participants were divided into two subgroups based on changes in viral loads between the baseline and follow-up visits: those who remained suppressed and those who became suppressed. All 55 participants who remained suppressed were on ART at baseline; 28 out of 53 PWH who became suppressed were off ART at baseline and started ART between the two visits.

The SIIA factor analysis yielded two factors: Factor1 loaded on CRP, D-dimer, and neopterin; Factor2 loaded on sTNFR-II. The NIIA factor analysis yielded two factors: Factor1 loaded on sTNFR-II and neopterin, and Factor2 loaded on MCP-1, sCD14, and IL6. The ND analysis yielded two factors: Factor1 loaded on Aβ42 and p tau; Factor2 loaded on NFL.

Figure 1 shows that higher levels of NIIA Factor1, reflecting increases in sTNFR-II and neopterin from baseline to follow-up, were associated with higher levels of ND Factor2, reflecting increases in NFL (r = 0.370, *p* = 0.0002). Because viral suppression is an important determinant of inflammation in PWH, we examined the interaction of CSF Immune Factor1 with viral suppression. In a multivariable model predicting CSF neurodegeneration Factor2 from viral suppression, CSF Immune Factor1, and their interaction, the interaction showed a trend towards significance (*p* = 0.0508); the main effect of CSF Immune Factor1 was significant (*p* = 0.0005), while that of viral suppression was not (*p* = ns). The correlation was much stronger for those who remained suppressed (r = 0.584, *p* = 8.43 × 10^−6^) than for those who became suppressed (r = 0.208, *p* = 0.157).

In the remained suppressed group, the proportion of females was higher than males (22.2% versus 5.8%, *p* = 0.023), as can be appreciated in Table 1.

The time between baseline and follow-up visits was not significantly related to any biomarker change factors (*ps* > 0.10).

Table 2 shows that older age was associated with higher levels of many biomarkers of inflammation and neurodegeneration at baseline, particularly CSF sTNFR-II (r = 0.438, *p* = 2.60 × 10^−6^) and CSF NFL (r = 0.374, *p* = 8.03 × 10^−5^). However, older age was not associated with *changes* in these same biomarkers.

Sex, ethnicity, and BMI did not significantly influence any of the CSF Immune or CSF Neurodegeneration change Factors (all *ps* > 0.20). At baseline, among the 83 participants who took antiretroviral, six different regimens were used. We classified the regimens into the following groups: non-nucleoside reverse transcriptase inhibitor (NNRTI)-based (*n* = 1), nucleoside reverse transcriptase inhibitor (NRTI)-based (*n* = 2), NNRTI/NRTI (*n* = 33), protease inhibitor-based PI/NRTI (*n* = 38), PI/NNRTI (*n* = 2), and three-class (*n* = 7). At follow-up, among the 107 participants who took antiretroviral, there were nine regimens used, including integrase inhibitor (II)-based/NRTI (*n* = 45), NNRTI-based (*n* = 1), NRTI-based (*n* = 2), NNRTI/NRTI (*n* = 23), PI/NNRTI (*n* = 1), PI/NRTI (*n* = 12), PI/II (*n* = 1), three-class (*n* = 21), and four-class (*n* = 1). There was no significant relationship between regimen type and any CSF Immune or CSF Neurodegeneration change Factors (all *ps* > 0.4). Nonetheless, we excluded groups with fewer than eight patients, and even this exclusion revealed no significant association between regimen type and changes in CSF Immune or CSF Neurodegeneration factors.

CSF Immune Factor1 was not related to any of the other neurodegeneration biomarkers (total tau, p tau, Aβ42, sAPPα, 8-oxo-DG). Those who remained virally suppressed in plasma had non-significant increases in CSF Immune Factor1 (Figure 2; 0.175 ± 0.743, *p* = 0.0946 versus the null hypothesis of no change), while those who became suppressed had significant decreases in CSF Immune Factor1 (−0.304 ± 0.790, *p* = 0.0098). The difference between the two subgroups was statistically significant (*p* = 0.0022). Baseline CSF sTNFR-II was not different in those who became suppressed (2.76 ± 0.246) from those who remained suppressed (2.79 ± 0.225, *p* = 0.598). CSF neopterin was lower in those with suppressed plasma HIV RNA at baseline than in those not suppressed (0.898 ± 0.244 versus 1.041 ± 0.234, *p* = 0.0034). Similarly, CSF neopterin at baseline was lower in those who had suppressed CSF HIV RNA < 50 c/mL than in those not suppressed at baseline (0.911 ± 0.227 versus 1.15 ± 0.237, *p* = 2.00 × 10^−5^).

Increases in plasma Immune Factor1 (CRP, D-dimer, and neopterin), correlated with increases in CSF Neurodegenerative Factor2 (NFL; r = 0.215, *p* = 0.0327). In a multivariable regression predicting CSF Factor2 from plasma Immune Factor1, in addition to change in detectability and their interaction, only the main effect of plasma Immune Factor1 was significant (*p* = 0.0426; ps for the other two terms in the regression were 0.0740 and 0.848, respectively).

## 4. Discussion

In this study, we analyzed selected soluble biomarkers associated with inflammation and neurodegeneration. We calculated changes in these biomarkers between the baseline and follow-up visits. We demonstrated that increases in neurodegenerative biomarkers in CSF, specifically NFL, were related to increases in inflammation and myeloid activation, as indexed by neopterin and sTNFR-II levels, respectively, in virologically suppressed PWH. Our findings were two robust considerations of potential confounds including demographics and HIV disease characteristics. This supports previous evidence that increases in inflammatory biomarker levels in plasma and CSF during HIV infection can expose neural cells to excessive and deleterious immune mediators [36,37,38]. Notably, chemokines are implicated in various neurological disorders. Although their primary function involves provoking immune responses by facilitating the precise movement of immune cells, they also exert direct influences on neuronal elements. Chemokines and their associated receptors stand as integral components orchestrating communication between neurons and inflammatory cells [39]. The correlation between change in inflammation markers and change in NFL was much stronger for those who remained suppressed than for those who became suppressed, suggesting that inflammation is more important as a driver of neurodegeneration in those who are durably virally suppressed than in those who have not yet achieved suppression. One potential explanation for this is that among unsuppressed PWH, viral replication itself is a more important driver of neurodegeneration than inflammation.

Consequently, the functioning and integrity of neural cells are affected, releasing the axonal marker NFL into the extracellular fluid and, ultimately, CSF.

The specific markers we found to be interrelated were sTNFR-II, neopterin, and NFL. This is concordant with previous studies showing neopterin to be elevated in PWH and in turn to strongly predict the progression of the disease [37]. We found that baseline CSF neopterin was significantly higher in those who were not suppressed at baseline than in those who were suppressed. This may explain why neopterin levels at follow-up decreased only in those who became suppressed—i.e., CSF neopterin in those already suppressed at baseline was already low (floor effect). sTNFR-II not only modulates the activity of TNF-α but also strongly correlates with HIV disease stage and progression. This dual role highlights its impact on immune regulation and clinical assessment, offering potential therapeutic intervention and disease monitoring avenues [40]. NFL is prominently expressed in large-caliber myelinated axons, with elevated cerebrospinal fluid (CSF) levels observed across various neurodegenerative disorders. This makes NFL a potential marker of axonal health and a candidate for diagnostic and prognostic applications in neurodegenerative diseases [27]. Activated microglia and macrophages drive neuroinflammation by releasing neurotoxins and inflammatory cytokines. This cascade of harmful molecules compromises neuronal integrity and function, emphasizing their crucial role in shaping neural health [41].

We did not find significant relationships between inflammation and markers of Alzheimer’s disease (AD) neuropathogenesis. AD is a profound neurodegenerative condition and the prevalent source of dementia linked to neurodegeneration among older individuals. This disorder is marked by a gradual decline in cognitive abilities, linked to the diminishment of synaptic and neuronal components and the presence of senile plaques and neurofibrillary tangles (NFT) in the brain [42]. There is concern for an increased risk of AD in the PWH [43,44]. HIV proteins such as Tat and gp120 modulate signaling and cellular pathways also impaired in AD, suggesting similarities and convergences of these two pathologies [45]. Our findings suggest that AD biomarkers are not linked to inflammation in HIV, though some previous studies have found AD biomarkers to show changes in PWH similar to those in AD [46,47,48]. A caveat is that the study participants were relatively young (mean age approximately 57 years) and therefore had a low risk of AD. Our results show that older age was associated with higher levels of many biomarkers of inflammation and neurodegeneration at baseline, particularly CSF sTNFR-II and CSF NFL. However, older age was not associated with changes in these same biomarkers.

A strength of this study is the long longitudinal follow-up and biomarker assessments at both baseline and follow-up. This helps us understand how things change over time, giving us a fuller picture. It is important to acknowledge that our study, while informative, does come with certain limitations. One notable aspect is that we did not assess all the possible biomarkers associated with inflammation and neurodegeneration. While we have made valuable strides in understanding these processes, there could be other biomarkers that we have not explored, and their inclusion might provide a more comprehensive view of the subject matter. We selected our biomarkers based on previous literature showing relationships between HIV disease and neuropathogenesis. Additionally, while most PWH are virally suppressed in modern cohorts, ours had substantial numbers of participants who were not suppressed at baseline. Nevertheless, all were suppressed at follow-up. Thus, our sample might not be representative of individuals with durable viral suppression.

Our study suggests that effectively treating neuroinflammation could potentially reduce neurodegeneration in virally suppressed PWH. Anti-inflammatory medications like TNF-α inhibitors might hold promise in preventing neurodegeneration by curbing inflammation. Further research and clinical trials are needed to confirm this possibility and explore the potential of anti-inflammatory interventions for preserving neural health in this population. Another question that remains unresolved is why PWH who have remained virally suppressed still have increased neuroinflammation. Indeed, we found that sustained viral suppression was associated with longitudinal decreases in neuroinflammation. There are several theories discussing probable mechanisms in continued immune activation in PWH undergoing ART. First, neuroinflammation increases in older people despite the lack of a frank immunosuppression [49]. Our study aligns with prior research, revealing that older individuals with HIV experience heightened neuroinflammation, systemic inflammation, and neurodegeneration compared to their younger counterparts. This underscores the complex interplay of age and these processes, emphasizing the need to further understand and address their implications for neural health in the HIV population [50,51,52]. Inflammation and neurodegeneration are linked together in other aging brain disorders such as Alzheimer’s disease (AD) [53,54,55,56]. Another hypothesis is HIV proteins generated even in the absence of viral replication activate immune responses [57]. Further investigations are essential to uncover the reasons behind increased neuroinflammation in virally suppressed PWH. Exploring these mechanisms could lead to valuable insights for therapeutic interventions and broaden our understanding of neuroinflammatory processes beyond HIV.

## 5. Conclusions

In conclusion, our study addresses the relationship between inflammation and neurodegeneration in virally suppressed people with HIV. We found that increased neuroinflammatory markers, including neopterin and sTNFR-II, were associated with elevated levels of neurodegenerative biomarkers, such as NFL, in virologically suppressed individuals. This underscores the potential impact of sustained inflammation on neural health even in the presence of viral suppression. While we did not observe significant connections between inflammation and Alzheimer’s disease markers in HIV, our findings emphasize the need for further exploration of the underlying mechanisms driving neuroinflammation in this population. The possibility of mitigating neurodegeneration through effective neuroinflammation treatment, possibly involving anti-inflammatory agents like TNF-α inhibitors, presents a promising avenue for future research and therapeutic interventions. Our results show older age was not associated with changes in these same biomarkers. As older individuals with HIV exhibit heightened neuroinflammation, systemic inflammation, and neurodegeneration, understanding these processes is crucial to developing strategies that preserve neural health and improve the well-being of people living with HIV. Further studies are warranted to unravel the complexities of neuroinflammation in virally suppressed individuals and its implications for neurological outcomes.

## Figures and Tables

**Figure 1 viruses-15-01835-f001:**
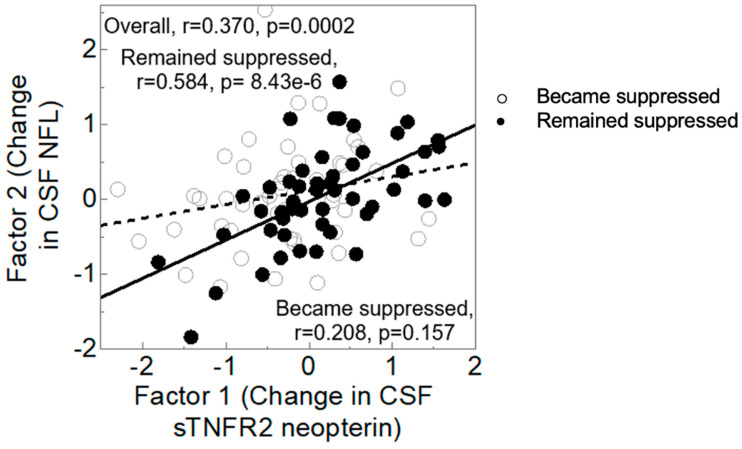
Correlation between CSF Immune Factor 1 (change in sTNFR2 and neopterin) and CSF Neurodegeneration Factor 2 (change in NFL). Filled circles, remained suppressed; open circles, became suppressed. Shaded regions represent 95% confidence bands for the regression fit.

**Figure 2 viruses-15-01835-f002:**
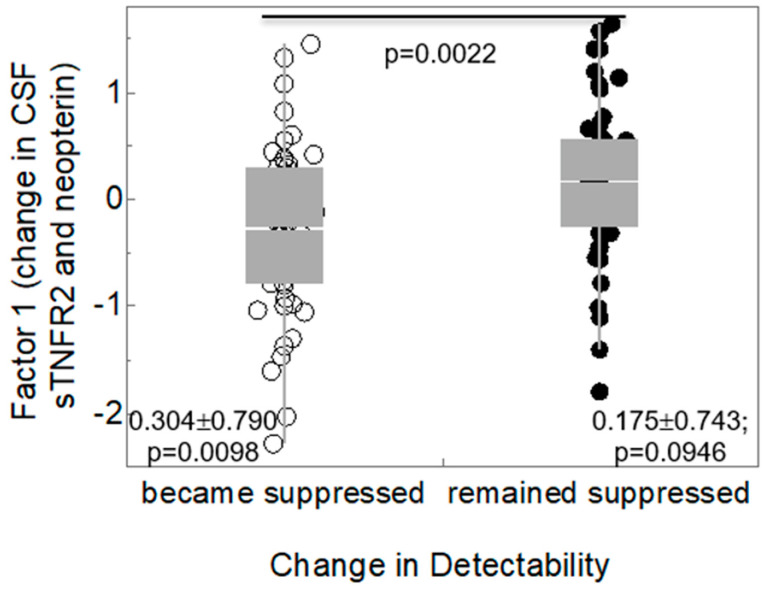
Box plot of viral detectability and CSF Factor 1. Those who remained virally suppressed in plasma had non-significant increase in CSF Immune Factor 1 (4 = 0.0946 versus the null hypothesis of no change), while those who became suppressed had significant decreases in CSF Immune Factor 1 (0.304 ± 0.790; *p* = 0.0098). The difference between the two subgroups was statistically significant.

**Table 1 viruses-15-01835-t001:** Participant demographics and clinical characteristics at baseline according to subsequent change in viral suppression status.

	Became Suppressed	Remained Suppressed	*p* Values
	53	55	--
Age—mean ± SD	55 ± 8.5	58 ± 7.9	0.151
Sex—N (%) female	3 (5.8)	12 (22.2)	0.023
Ethnicity—N (%)			0.0874
Black	17 (41.5)	24 (58.5)	--
Hispanic	5 (50)	5 (50)	--
Non-Hispanic white	30 (57.7)	22 (42.3)	--
Other	0	3 (100)	--
Current CD4—median (IQR)	579 (374–932)	579 (358–784)	0.618
CD4 nadir—median (IQR)	109 (26–237)	67 (11–184)	0.633

**Table 2 viruses-15-01835-t002:** Older age was associated with higher levels of some biomarkers of inflammation and neurodegeneration, but not with changes in these biomarkers. Significant values are in bold.

	Biomarker at Baseline		Biomarker Change	
	r	*p*	r	*p*
Log_10_ plasma 8-oxo-dG	0.124	0.205	−0.0214	0.828
Log_10_ plasma AB-42	−0.040	0.690	−0.0531	0.589
Log_10_ plasma CRP	−0.120	0.228	0.0131	0.894
**Log_10_ plasma D-dimer**	**0.277**	**0.004**	0.0744	0.448
Log_10_ plasma IL-6	−5 × 10^−3^	0.961	−0.0412	0.675
Log_10_ plasma MCP-1	0.088	0.370	0.0009	0.993
Log_10_ plasma neopterin	0.111	0.258	0.0023	0.981
**Log_10_ plasma sAPPα**	**0.193**	**0.047**	0.0211	0.831
Log_10_ plasma sCD14 Plasma	0.010	0.917	0.148	0.130
Log_10_ plasma sCD40L Plasma	0.111	0.256	−0.0469	0.635
Log_10_ plasma sTNFR-II Plasma	0.077	0.436	−0.0477	0.627
Log_10_ CSF 8-oxo-dG	0.129	0.188	0.110	0.263
Log_10_ CSF AB-42	0.075	0.444	0.1708	0.0815
Log_10_ CSF IL-6	0.122	0.212	0.0485	0.622
Log_10_ CSF MCP-1	0.161	0.100	−0.0042	0.966
**Log_10_ CSF** Neopterin	**0.228**	**0.019**	0.0993	0.328
**Log_10_ CSF NFL**	**0.374**	**8.03 × 10^−5^**	0.0488	0.619
Log_10_ CSF sAPPa	−0.060	0.558	−0.1888	0.0627
Log_10_ CSF sCD14	0.022	0.826	−0.0543	0.581
**Log_10_ CSF sTNFR-II**	**0.438**	**2.60 × 10^−6^**	0.187	0.0565
Log_10_ CSF tau pT181	0.164	0.092	0.128	0.208
**Log_10_ CSF Total tau**	**0.278**	**0.004**	0.254	0.0109

## Data Availability

The datasets generated and/or analyzed during the current study are available in the CHARTER repository upon reasonable request.

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
