# Peer review of "Increasing Neuroinflammation Relates to Increasing Neurodegeneration in People with HIV"

_viruses, 2023, doi:10.3390/v15091835_

Round 1

Reviewer 1 Report

Manuscript entitled " Increasing Neuroinflammation relates to Increasing Neurodegeneration in PWH”.

Dear Editor,
I have completed my review of the aforementioned manuscript. Here's a concise summary of my observations and recommendations:
1. Scope and Originality: The paper explores a longitudinal study,
after 12 years using standardized evaluations, 14 biomarkers of neuroinflammation and immune activation and systemic inflammation and neurodegeneration in plasma and CSF at two-time points and calculated changes. The increases in neuroinflammation and immune activation and systemic inflammation and immune activation in PWH were associated with corresponding increases in ND, suggesting that reducing neuro/systemic inflammation might slow or reverse neurodegeneration.
2. Methodology: The methodologies used
is well organized, and the experiments are well performed.
3. Results and Interpretation: The data presented is comprehensive, but there is a need for better visualization tools. Graphs and tables are somewhat crowded.
The statistical analysis is well performed.

4. Discussion: Data interpretation is well discussed, and the message is clearly stated.
5. Literature Review:
The literature cited is relevant.
6. Conclusion: The conclusions are pertinent,
and the length is commensurate with the message, given the novel nature of this study.
In summary, the manuscript presents potentially groundbreaking findings, it needs refinement in its presentation. With some modifications, it can contribute to addressing
the relationship between inflammation and neurodegeneration in virally suppressed people with HIV.

For the mentioned reasons, the manuscript may be accepted for publication with minor revision.

Reviewer 2 Report

Increasing Neuroinflammation relates to Increasing Neuro-degeneration in PWH by Azin Tavasoli

Dear Editor-in-Chief (Pathogens, MDPI)

Basically, this study evaluates soluble biomarkers associated with several inflammation and neurodegeneration among supressed and non supressed HIV-1 seropositive patients. They calculated changes in these biomarkers between the baseline and follow-up visits and these authors have found increases in certain neurodegeneration biomarkers in CSF, specifically NFL, were related to increases in inflammation and 248 myeloid activation, as indexed by neopterin and sTNFR-II levels, respectively, in virolog- 249 ically suppressed PWHlthough some of authors are very important in the clinical field of HIV-1, this manuscript contains some imprecise concepts in terms of neurogeneration, which must be solved before its final acceptation. For examplel, they indicated in line 248 ¨……myeloid activation, as indexed by neopterin and sTNFR-II levels, respectively, in virologically suppressed PWH. However, neopterin is a general marker of neurodegeneration but does not reflect mieloid activation. For example, CX3CR1/fractalkine can be associated with mieloid/microglial activation.

Please, indicate the sensivity of each marker for ELISA/inmmunoassay technic for these 14 analyzed marker and also describe this biochemical thecnic in material and methods.

Remove grups of ART with n=2 and n=7 in order to avoid methodological errors here.

The discusion should be improved following these suggestions. Please, include the posible effect of age of these markes in these seropositive patients within the conclusion section.

My Decision is minnor revision. Please, take into account my suggestions

Commments to the Author and Editor-in-Chief

Major comments

Markers such as Phosphorylated tau (p-tau) and Ab42 are associated with age-related dementia but are less evidente in HIV-cognitive impairment. Thus, the effect of age should be discussed better in the discusión and also included in the conclusion.

Minnor comments

The meaning of the abreviature PWH is not present in the manuscript

-Line 58. Please, indicate chemokine (chemotactic cytokines) here.¨HIV infection triggers an inflammatory response that disrupts the equilibrium of chemotactic cytokines in both plasma and cerebrospinal fluid (CSF). This alteration affects immune cell migration and communication [8-11]

-Line 59. This sentence is true but their manuscript did not tested these inflammatory markers in neurons (in vitro or postmorten brains for example). Their study was focused on systemic inflammatory and CSF markers without a real study with neurons. Please, take into account this comment.

Line 64. Please, their study does not highlight the potential role of chemokines on neuronal signaling and repair and also signaling cascades, synaptic plasticity, and neurotransmitter reléase were not tested in this study. Thus, remove or redone this parragran given this clinical orientation with inflammatory mediators in both group of patients here.

Thus, redone this parragrah that must be focus on the clinical aim for evaluating a possible differential changes on inflammatory mediators in these seropositive patients. However, they don`t evaluate the role of inflammatory makers in neuronal los and the lack of synaptic plasticity can not be considered here. Thus, redone or remove this sentence from line 62-69 ¨Moreover, they possess the ability to impact the functionality and structural integrity of various cell types, including neurons, extending their influence beyond the immune domain. This dual role highlights the potential for cytokines to affect immune-neural interactions [12]. Since most neural cells express cytokine receptors, cytokines can affect neuronal signaling and repair, potentially disrupting cascades that underlie information processing, synaptic plasticity, and neurotransmitter release. Furthermore, the ability of cytokines to elicit cellular responses within neurons might either exacerbate or attenuate the reparative processes following neural injury or insult [11]

Did you find infiltrated monocytes-CCR5 positive fom the circulation into the brain? Why there are absence of changes on markers of monocyte activation levels (ie: soluble CD14) among these seropositive patients? This is strange for me and we must explain a possible reason for these observed clinical discrepances in terms of macrophage activation, including the lack of effect for MCP-1. Is there any relationshipp betwen CD4 nadir and this differential expresión on NF-L, TNFR-II or Tau levels among both groups of HIV-1 infected patients? I miss the inclusion of antiinflammatory mediators as better index of inflammation in this study. For example, it is more appropiate the evaluate the rate IL-4/IL-6 as index of inflammation.

-Line 85-86. Please, don`t talk about neurocognitive impairment [18, 19]. It is cognitive impairment; it is not posible to talk on neurodegeneration without the contribution of glial cells (microglia, astrocytes) as well as the vascular endothelium. There is a real communication betwen neurons and glial cells in the brain.

Line 88 …. ¨neopterin, a product of the guanosine triphosphate pathway, is synthesized in monocyte/macrophage cells. It serves as a marker of immune activation, offering insights into cellular dynamics¨. This is true and correct but neuopterin is a general markers of inflammation and there are better markers for the evaluation of inflammation.

Please, detail this sentence (line 96-98), which is not clear for me.

¨cART decreases CSF neopterin; however, despite suppression of plasma and CSF HIV RNA to CSF neopterin often remains mildly elevated, raising the important question of whether neopterin elevation is caused by continued CNS infection or persistent CNS injury [21]¨, If CNS infection is persistent, explain the absence of changes for MCP-1 among both HIV-1 study groups in your study. Neopterin is not the best and accurate marker of neuriinflammation

Line 103-115. Please, discuss the interactive effect of HIV-1 infection and aging on soluble tumor necrosis factor receptor II (sTNFR-II) and the effect of viral load supression in your study. In other words, how does age affect soluble tumor necrosis factor receptor II (sTNFR-II) levels among seropositive patients in this study? Is there any relationship betwen CD4 nadir, neurodegeneration and NF-L or TNFR_II levels in your study?

Line 119. ¨The most frequently used biomarker of neurodegeneration is the neuro- 119 filament light protein (NFL)¨ . As expert in neurogeneration and also HIV neuropathogenesis, there are better markers of neuronal but NF-L is valid as marker of axonal damage. I would expect to analyze some marker of apoptosis (Apaf-1/caspase-3, cytocrome c, etc).

Line 122. In PWH, CSF NFL levels are negatively correlated with blood CD4+ T lymphocyte counts, demonstrating the relationship 123 between neuronal injury and systemic HIV infection [33]. Are you talking about CD+ nadir?

Line 128. ¨We measured changes in biomarker levels from baseline to follow-up over 12 years, hypothesizing that increases in inflammation would correlate with increases in neurodegeneration biomarkers¨. Maybe, this correlation take place earlier but your data can not explain it.

Line 130. ¨We also predicted that viral suppression between the baseline and follow-up visits would be associated with decreasing markers of neuroinflammation and immune activation (NIIA) and systemic inflammation and immune activation (SIIA)¨. This is obvious¡

Line 140. ¨Participants with severely confounding medical and neuropsychiatric conditions such as active psychosis were excluded¨. Please, provide published findings on psychosis and HIV-1 cognitive impairment in terms of inflammatory mediators here. Please, include a reference that demonstrate this feature here.

Line 149. ¨CD4+ T-cell count was measured in Clinical  Laboratory Improvement Amendments (CLIA)–certified laboratories at each site¨. Please, detail the technical procedure for CD4 analysis as well as immunoassay protocol in material and methods.

Line 154-156. ¨Please, indicate sensitiviy of ELISA assay or inmmunoassay for all these described markers. and data from a panel of 14 soluble biomarkers measured by immunoassay was available at both assessments: interleukin 6 (IL-6; CSF and plasma), soluble tumor necrosis factor type II (sTNFR- 156 II; CSF and plasma), neopterin (CSF and plasma), monocyte chemoattractant protein type 157 1 (MCP-1; CSF and plasma), soluble CD14 (sCD14; CSF and plasma), and 8-Oxo-2'-deox-yguanosine (8-oxo-DG - a marker of nucleic acid oxidation; CSF and plasma), neurofila ment light (NFL; CSF only), total Tau (CSF only), phosphorylated tau (CSF only), soluble amyloid precursor protein (sAPP)-a (CSF and plasma) and amyloid b-42 (Ab42; CSF only), soluble CD40 ligand (sCD40L, plasma only), C-reactive protein (CRP; plasma only), and D-dimer (plasma only).

Please, indicate the sensivity of each marker for ELISA/inmmunoassay technic.

In addition, these are age-related dementia markers (total Tau (CSF only), phosphorylated tau (CSF only), soluble amyloid precursor protein (sAPP)-a (CSF and plasma) and amyloid b-42 (Ab42; CSF) although is overexpressed in seropositive postmorten brains .

Line 174. -Describe these possible confounds factors in the discussion, including demographics and indicators of HIV disease status by using multiple regression.

Line 188. ¨ The SIIA factor analysis yielded two factors: Factor1 loaded on CRP, d-dimer, and 188 Neopterin¨. CRP is not a real marker of inflammation and d_Dimer reflects vascular problems while neuropterin is a general marker of inflammation. There are better and specific markers of inflamamtion for detection inflammation and neurodegeneration in HIV-1 patients.

Line 198. ¨The interaction showed a trend towards significance (p=0.0508); True, this is almost signifficative. However, this reviewer does not consider a correlation at the end of sentence in line 199. ¨ The correlation was much stronger for those who remained suppressed (r=0.584, p=8.43e- 201 6) than for those who became suppressed (r=0.208, p=0.157)¨. In fact, the correlation is not signifficant and very low in case of suppressed (r=0.208, p=0.157); please, add n.s (without significative effect here). In addition, data showed in figure 1 could be influenced by the low percentage of males (only 5.8 %). In fact, the remained suppressed group, the proportion of females was higher than males 204 (22.2% versus 5.8%, p=0.023). Although this effect is signifficative, explain the sexodimorphic effect found among males and females here.Is there any hormonal regulation among these markers in seropositive patients?

Table-2. The correlation for Abeta-42 is very low and D-dimer could only reflect cardiovascular alterations at the baseline level in this patients. What does the conexión between neurode generation and higher D-dimer levels in HIV-1 seropositive patients? I am not confident with the real stimation plasma sAPP levels as marker of neurodegeneration. In fact, the analisis of correlation is poor without statistical effect in this table (r=0.193, p=0.047); although is signifficant at baseline levels, the correlation is 0.19 (low) and these correlative effect dissapaer. In additon, explain the reason by the lack of effect for MCP-1 levels among HIV-1 seropositive patients? If HIV-1 is persistent in the brain, I would expect MCP-1 increases (at least at the systemic level ) at baseline). The analsysis of this table reflect signifficative effects for neopterin, TNFR-II, NF¨and Tau (NF-L). However, these correlations are low. The high r is for TNFR-II (r=0.43 with a very strong signifficative effect). Also explain better the correlation for Tau at baseline and biomarker change in the discusión. Are these changes on total Tau or phospho-tau levels? (Log10 CSF Total Tau r=0.278, p= 0.004 at baseline vs r=0.254, p=0.0109). So, are Tau levels really affected in your study? Are NF-L increases reflect the cognitive impairment or are consecuence of axonal damage without dementia? Which is the contribution of synaptic prunnig in your study among both HIV-1 study groups? In my  opinion, we can not talk on mielodi cells here because we have not evaluated markers here. Although neurons are CD4 negative, there is controversy about how HIv-1 infects neurons (or not). Please, describe this feature in the discusion and explain the absence of changes on CD4 nadir for these seropositive patients in terms of neurodeneration. Also, discuss how CD4 nadir could affect or not your evaluated markes in the discusion and also explain better the interactive  effect of aging since Alzheimer or dementia markers are not upregulated in your study.

Line 216. ¨ The classified the regimens into 216 the following groups: non-nucleoside reverse transcriptase inhibitor (NNRTI)-based 217 (n=1), nucleoside reverse transcriptase inhibitor (NRTI)-based (n=2), NNRTI/NRTI (n=33), 218 protease inhibitor-based PI/NRTI (n=38), PI/NNRTI (n=2), and 3 class (n=7).¨. Is really worthy to include ART treatment with n=2 or n=7 in these ART groups? Please, increase this size sample for robust conclusions with these ART régimen of treatment. I would suggest to remove this groups with small number of patients (n=2 and n=7), which prevent methodological errors.

Line 223. ¨There was no significant relationship between regimen type and any CSF Immune or CSF Neurodegeneration change Factors (all ps>0.4)¨. Please, indicate these groups. If belong to n=2 and n=7 samples this could be the cause of lack of effect by this low size sample.Obvious¡

Line 226.Please, explain how you are able to differenciate bewteen tTau, pTau by ELISA. Do you have a ELISA or immunoassay able to discrimante betwen both of them? This is strange for me.

Line 252. ¨ Chemokines play an important role in the nervous system, encompassing crucial functions such as guiding the migration of neurons, promoting cellular growth, modulating synaptic activity, and orchestrating neuroinflammatory responses¨. Please, only describe MCP-1 because we only indicated this chemokine in your study. The functions of chemokines are out of the aim of this study. Please, remove this sentence from 252 until 254 line (¨Chemokines play an important role in the nervous system, encompassing crucial functions such as guiding the migration of neurons, promoting cellular growth, modulating synaptic activity, and orchestrating neuroinflammatory responses¨.

Line 260. Please, include a justification with published papers that corroborate that inflammation is the cause of NF-L increases in HIV-1 seropositive patients. Otherwhise, this is only a hypothesis that must be confirm here.

Line 281. If you don`t show molecular evidences on increased blood-brain barrier (BBB) permeability in your study, then remove this sentence from   line 281 until 291. ¨ Increased blood-brain barrier (BBB) permeability is linked to reactive astrocytes and the potential release of viral proteins. This interplay underscores  the complex communication between neural and vascular elements, emphasizing the im-portance of BBB integrity in neural health and disease [42-45]. Altered BBB permeability leads to a permissive environment for neuronal injury and death. The breached barrier allows entry of potentially harmful agents, leading to neuronal damage and dysfunction.This disruption highlights the critical role of BBB integrity in maintaining neural health and underscores the need for interventions to restore barrier function and mitigate neuronal compromise [46]. Activated microglia and macrophages drive neuroinflammation by releasing neurotoxins and inflammatory cytokines. This cascade of harmful molecules compromises neuronal integrity and function, emphasizing their crucial role in shaping neural health [46]. In fact, NF-´L increases in your study, although without analisis of molecular markers able to detect this BBA permebilization among HIV-1 infected patients.

Line 293. ¨We did not find significant relationships between inflammation and markers of Alz-  heimer's disease (AD) neuropathogenesis¨. Please, revise the age of patients. Maybe, these is the cause and could occur at later age.

Line 301. ¨Our findings suggest that AD biomarkers are not linked to inflammation in HIV, though some previous studies have found AD biomarkers to show changes in PWH similar to those in AD [51-53]. Maybe, the reason of this discrepance is age dependent. In fact, AD is a age-related dementia, which is agravated by neuroinflammation. In fact, we suggest this effect (¨A caveat is that the study participants were relatively young (mean age approximately 57 years) and therefore had a low risk of AD¨).

Line 308. ¨One notable aspect is that we didn't assess all the possible biomarkers associated with inflammation and neurodegeneration¨. I told you before. In fact, there are better markers of inflammation. In my opinion, index as Il-4/Il-6 ratio or IL-10/IL-6, IL-1 beta/IL-4 reflects the imbalance betwen proinflammatoy cytokine release and antiinflamatory mechanisms here.

Line 319. ¨… Anti-inflammatory medications like TNF-α inhibitors might hold promise in preventing neurodegeneration by curbing inflammation¨. Is there a clinical trial with these TNF Alpha antagonist as occur with anakinra (a TNF antibody that blocks TNF alfa)

Conclusion. Please, include the posible effect of age on these inflammatory and neurodegenerative markes in these seropositive patients.
